# A Retrospective Cohort Study of Safety Outcomes in New Zealand Infants Exposed to Tdap Vaccine in Utero

**DOI:** 10.3390/vaccines7040147

**Published:** 2019-10-11

**Authors:** Helen Petousis-Harris, Yannan Jiang, Lennex Yu, Donna Watson, Tony Walls, Nikki Turner, Anna S. Howe, Jennifer B. Griffin

**Affiliations:** 1Vaccine Datalink and Research Group, Department of General Practice & Primary Healthcare, School of Population Health, Faculty of Medical and Health Sciences, University of Auckland, Private Bag 92019, Auckland 1142, New Zealand; a.howe@auckland.ac.nz; 2Department of Statistics, University of Auckland, Private Bag 92019, Auckland 1142, New Zealand; y.jiang@auckland.ac.nz (Y.J.); lennex.yu@pancohc.com.tw (L.Y.); 3Immunisation Advisory Centre, Department of General Practice & Primary Healthcare, School of Population Health, Faculty of Medical and Health Sciences, University of Auckland, Private Bag 92019, Auckland 1142, New Zealand; d.watson@auckland.ac.nz (D.W.); n.turner@auckland.ac.nz (N.T.); 4Department of Paediatrics, University of Otago, PO Box 4345, Christchurch Mail Centre 8140, New Zealand; tony.walls@otago.ac.nz; 5Social, Statistical, and Environmental Sciences, RTI International, 3040 East Cornwallis Road, Research Triangle Park, NC 27709-2194, USA; jenngriffin@rti.org

**Keywords:** post-marketing surveillance, vaccine safety, pertussis, Tdap, pregnancy, infant

## Abstract

We aimed to evaluate the safety of maternal Tdap; thus, we assessed health events by examining the difference in birth and hospital-related outcomes of infants with and without fetal exposure to Tdap. This was a retrospective cohort study using linked administrative datasets. The study population were all live-born infants in New Zealand (NZ) weighing at least 400 g at delivery and born to women who were eligible for the government funded, national-level vaccination program in 2013. Infants were followed from birth up to one year of age. There were a total of 69,389 eligible infants in the cohort. Of these, 8299 infants were born to 8178 mothers exposed to Tdap (12%), primarily between 28 and 38 weeks gestation as per the national schedule. Among the outcomes, we found a reduced risk for moderate to late preterm birth, low birth weight, small for gestational age, large for gestational age, respiratory distress syndrome, transient tachypnea of newborn, tachycardia or bradycardia, haemolytic diseases, other neonatal jaundice, anaemia, syndrome of infant of mother with gestational diabetes, and hypoglycemia in infants born to vaccinated mothers. There was no association between maternal Tdap, infant Apgar score at 5 min after birth, asphyxia, sepsis or infection, or hypoxic ischemic encephalopathy. Infant exposure to Tdap during pregnancy was associated with a higher mean birthweight (not clinically significant) and higher odds for ankyloglossia and neonatal erythema toxicum diagnoses. There were insufficient observations to allow examination of the effect of Tdap on extreme preterm and very preterm birth, and stillbirth, infant death, or microcephaly. Overall, we found no outcomes of concern associated with the administration of Tdap during pregnancy. NZ Health and Disability Ethics Committee Approval #14/N.T.A/169/AM05.

## 1. Introduction

Pertussis vaccination programmes have had a dramatic impact on pertussis morbidity and mortality, particularly for infants. The burden of severe morbidity and mortality now falls primarily on infants too young to be vaccinated. However, a resurgence in disease is being observed in many countries using acellular vaccines [1] and, to a lesser extent, countries using whole-cell vaccines [2].

Natural immunity to pertussis varies in terms of completeness and duration. Furthermore, immunity via current acellular vaccines, while preventing clinical disease [3], does not prevent carriage or transmission [4,5]. These issues pose challenges for the control of pertussis.

Maternal immunisation as a strategy to prevent neonatal and infant mortality has been well illustrated with the success of the World Health Organization (WHO)/UNICEF neonatal tetanus elimination programme in low-income settings. Following the implementation of maternal tetanus immunisation programmes in at-risk populations, mortality from neonatal tetanus declined by 94% (95% CI [80, 98]) [6]. Evidence for the effectiveness of maternal influenza vaccination in preventing influenza for the first months of life has also supported the move to a maternal vaccination approach [7]. Since 2011, some countries, such as the UK, have begun maternal pertussis immunisation [8,9] and the strategy has proved highly effective [10,11,12,13]. While there are no theoretical safety concerns about using inactive or subunit vaccines in pregnant women [14], there were few empirical data available during the early years of these programmes [8].

Between 2011 and 2013, New Zealand (NZ) experienced the largest pertussis epidemic since 2000. The number of notified cases of pertussis rose dramatically from July 2011 and remained high throughout 2012 and 2013, with rates of over 270/100,000 infants under one year of age. Among notified cases in the less than six weeks of age group, 56% were hospitalised, with 23% of these requiring multiple hospitalisations. Because of this disease burden, a booster dose of acellular pertussis vaccine was recommended in 2012 and then funded in 2013 for women between 28 and 38 weeks gestation.

We previously reported the maternal outcomes from this study [15]; here, we report infant outcomes by examining the difference in birth and hospital-related outcomes of infants with and without fetal exposure to Tdap.

## 2. Methods

### 2.1. Study Population and Variables

The study population included all live-born infants in NZ weighing at least 400 g at delivery and born to women who were eligible for the NZ Ministry of Health (MoH)-funded, national-level vaccination program in 2013 (that is, all pregnant women in NZ between 28 and 38 weeks gestation). Infants were followed from birth up to one year of age (Figure 1).

The independent binary variable was exposure to Tdap during the mother’s pregnancy. Analysis takes in to account all Tdap vaccinations during pregnancy with 5% occurring outside of the 28–38 gestation window.

### 2.2. Study Outcomes

The study outcomes were prioritized according to the categories presented in the assessment of vaccine safety in pregnant women, as defined by WHO and the Brighton Collaboration task force, and termed ‘priority outcomes’, ‘outcomes’, and ‘suggested outcomes’ [16]. We used these outcomes as a guide and linked them to International Classification of Disease 10, Australian Modification (ICD-10-AM) codes from relevant chapters A, B, E, F, G, J, P, Q, R and Z to identify outcomes potentially associated with exposure to maternal Tdap vaccination.

Each outcome variable is dichotomous, with possible values of ‘yes’ or ‘no’. Where an infant experienced the same outcome on multiple occasions during the study period, only the first episode was considered. Priority outcomes were stillbirth, perinatal death, neonatal death, infant death, preterm birth, small for gestational age (SGA), congenital anomalies (major and minor), asphyxia, infection, and sudden infant death syndrome. Other outcomes with significant findings are also reported. We included all codes across 99 possible diagnosis fields except the Q-codes (congenital anomalies) where only the primary codes were used.

Additional covariates include demographic and clinical characteristics and the model of care variable (midwife, obstetrician, general practitioner).

### 2.3. Data Sources

Our data sources for this study have been previously described in detail [15]. They consisted of the National Health Index Database of demographic information; National Minimum Data Set of all hospital discharges in NZ following inpatient episodes of care; Mortality Data Set of underlying causes of all deaths registered in NZ, including fetal deaths (stillbirths); National Maternity Collection of data on primary maternity services and inpatient and day-patient health event data from nine months before and three months after a birth for mothers and infants; and the Immunisation Subsidies Collection of data on the fee-for-service payments made to general practitioners for providing government-funded immunisations.

### 2.4. Statistical Methods

For all infants, follow-up began at birth and infants were censored at the first event outcome of death, first birthday, or loss to follow-up (no record in any of the data sources).

Demographics and clinical characteristics of infants and mothers were first summarised descriptively, overall and by infants who did and did not have fetal exposure to Tdap. Continuous variables were described as mean, standard deviation (SD) and median and inter-quartile range (Q1, Q3). Categorical variables were described as frequency and percentage.

Each reported outcome (with at least one event) was next described quantitatively with frequencies and incidence rates, for exposed and unexposed infant groups separately. The median and interquartile range (Q1, Q3) of infants’ age at the time of each outcome was also reported.

The relationship between fetal exposure to Tdap and infant outcomes were investigated using adjusted regression models appropriate to the distribution of outcome. Adjusted regression analyses accounted for pre-defined confounding variables and were used to support the main conclusions. Each model was adjusted for birth status (single live birth, other birth); maternal ethnicity (Māori, Pacific, Asian, NZ European or other); NZ Deprivation Index 2013 (deciles grouped into quintiles); maternal age (in years); history of antenatal care (total number of lead maternity carer visits); maternal body mass index (kg/m^2^); history of chronic disease (yes, no); parity (0, 1+); model of care (District Health Board (DHB), midwife, obstetrician/general practitioner, no lead maternity carer/other); and influenza vaccination (yes, no) during the same pregnancy. Outcomes were excluded if the proportion of events was <0.1% in either exposed or unexposed group, or the number of events in the exposed group was <10.

Continuous outcomes (birthweight and Apgar score at 5 min after birth) were analysed using linear regression models. The effect of fetal Tdap exposure was estimated as mean difference with 95% confidence intervals. Those outcomes diagnosed at delivery with no follow-up time were considered as a binary variable and analysed using logistic regression models. Odds ratios (ORs) and 95% confidence intervals (CIs) were reported accordingly. An OR of <1 indicated lower odds of having the outcome with fetal exposure to Tdap and was statistically significant if the CI didn’t include 1.

Statistical analysis was performed using SAS version 9.4 (SAS Institute Inc., Cary, NC, USA). All statistical tests were two-sided at 5% significance level (*p* < 0.05).

## 3. Results

### 3.1. Study Cohort

There were a total of 69,389 eligible infants in the cohort. Of these, 8299 infants were born to 8178 mothers exposed to Tdap (12%) (Figure 1).

Of infants born to women eligible to receive the vaccine, 51.2% were male. Infants of European ethnicity comprised 67.0% of the vaccine-exposed group, while infants of Māori ethnicity comprised 13.2%. The deprivation quintile of exposed infants ranged between 20.4% and 21.7% for the first four deprivation quintiles. The most deprived quintile contributed 15.2% of exposed infants (Table 1).

The effect of maternal Tdap on hospital-related infant outcomes diagnosed at birth, on eligible maternities, by Tdap exposure are summarised in Table 2 and Table 3.

### 3.2. Events of Delivery

There were insufficient number of cases in the vaccine-exposed group to assess the association between Tdap exposure and stillbirth (*n* = 9), extreme (*n* = 0) and very preterm birth (*n* = 9), and extreme (*n* = 0) and very low birth weight (*n* = 9).

We found a reduced risk associated with exposure to vaccine for moderate to late preterm birth (OR 0.83; 95% CI [0.73, 0.95]).

### 3.3. Physical Examination and Anthropometric Measurements

There were insufficient observations available to allow examination of the effect of Tdap on extreme low birth weight (LBW) and very LBW.

There was no mean difference in Apgar score between vaccine-exposed and unexposed groups (Table 3). A small but significantly higher mean birthweight was observed in the vaccine-exposed group with a mean difference of 35.59 g (95% CI [21.39, 49.78]).

We found reduced risks in the Tdap exposure for LBW (OR = 0.78; 95% CI [0.65, 0.94]), SGA (OR = 0.72; 95% CI [0.57, 0.91]), and large for gestational age (LGA) (OR = 0.57; 95% CI [0.36, 0.89]).

### 3.4. Congenital Anomalies

Two congenital anomalies that had enough cases to include in the regression models were deformities of feet and ankyloglossia (tongue-tie). There was no association with deformities of feet (OR = 0.963; 95% CI [0.61, 1.52]). There was an increased odds associated with ankyloglossia (OR = 1.241; 95% CI [1.04, 1.47]). There were three infants with microcephaly, none was born to mothers exposed to Tdap. There were insufficient observations in both the exposed and unexposed groups to explore other congenital anomalies

### 3.5. Neonatal Conditions Classified by Organ System

There were insufficient observations available to allow examination of the effect of Tdap on neonatal sepsis due to Streptococcus, group B, other and unspecified streptococci, *Staphylococcus aureus*, other and unspecified staphylococci, *Escherichia coli*, anaerobes, congenital viral infections, congenital infectious and parasitic diseases, neonatal infective mastitis, and neonatal urinary tract infection or infection specific to the perinatal period, specified or unspecified.

We found no association between exposure to vaccine and asphyxia, specified or unspecified sepsis, candidiasis, omphalitis, neonatal conjunctivitis and dacryocystitis, neonatal skin infection, or hypoxic ischemic encephalopathy. We found a reduced risk among theTdap vaccine exposure group for respiratory distress syndrome (OR = 0.65; 95% CI [0.52, 0.81]), transient tachypnea of newborn (OR = 0.84; 95% CI [0.72, 0.98]), tachycardia or bradycardia (OR = 0.69; 95% CI [0.50,0.95]), haemolytic diseases (OR= 0.66; 95% CI [0.44, 0.99]), other neonatal jaundice (OR = 0.87; 95% CI [0.76, 0.10]), syndrome of infant of mother with gestational diabetes (OR = 0.68; 95% CI [0.49, 0.96]), and hypoglycaemia OR = 0.80; 95% CI [0.68, 0.93]).

There was one other infant outcome not elsewhere described that was significantly associated with exposure to Tdap, neonatal erythema toxicum. After adjustment, the association remained significant (OR = 1.66; 95% CI [1.16, 2.37]).

### 3.6. Infant Death

There were insufficient events (*n* = 4) in the exposed group to allow examination of the effect of Tdap on infant death.

## 4. Discussion

This study sought to examine the safety for the infant after their mothers received Tdap during the pregnancy. We examined the difference in rates of key outcomes between those infants exposed and not exposed. Among the outcomes, we found a reduced risk for moderate to late preterm birth, LBW, SGA, LGA, respiratory distress syndrome, transient tachypnea of newborn, tachycardia or bradycardia, haemolytic diseases, other neonatal jaundice, anaemia, syndrome of infant of mother with gestational diabetes, and hypoglycemia in infants born to vaccinated mothers. There was no association between maternal Tdap and infant Apgar score at 5 min after birth, asphyxia, sepsis or infection, or hypoxic ischemic encephalopathy. Infant exposure to Tdap during pregnancy was associated with a higher mean birthweight (not clinically significant) and higher odds for ankyloglossia and neonatal erythema toxicum diagnoses. There were insufficient observations to allow examination of the effect of Tdap on extreme preterm and very preterm birth, stillbirth, microcephaly, and infant death. Overall, we found no outcomes of concern associated with the administration of Tdap during pregnancy.

## 5. Interpretation

Since the implementation of maternal Tdap programmes internationally there has been limited data on infants beyond birth outcomes. Most recently, a Vaccine Safety Datalink (VSD) study assessed 413,034 live births from 2004 to 2014 for the risk of hospitalisation and showed no overall increased risk (adjusted OR = 0.94; 95% CI [0.88, 1.01]) or death (adjusted OR = 0.44; 95% CI [0.17, 1.13]) in the first six months of life associated with maternal pertussis [17]. While we did not measure overall hospitalisation, our findings support this study.

While we had insufficient observations for infants born extremely preterm or very preterm, we found no increased risk for moderate to late preterm birth. The previous evidence regarding the relationship between Tdap vaccination and preterm birth is mixed. Several VSD studies have found no relationship between Tdap vaccination during pregnancy and preterm birth [18,19]. The Texas-based study reported a non-significant trend towards a protective effect against preterm birth (>37 weeks) with an adjusted OR for preterm delivery of 0.68 (95% CI [0.45, 1.03]) [20], and a retrospective cohort study found infants from unvaccinated mothers were more likely to be born preterm (<37 weeks), 6% compared with 12% (*p* = 0.001) [21].

While we had insufficient observations available to allow examination of the effect of Tdap on extreme LBW and very LBW, we found a reduced risk for LBW associated with exposure to Tdap in pregnancy. In the Texas retrospective record review study, the adjusted OR for LBW was 0.76 (0.51–1.14) and for very LBW was 0.24 (0.05–1.20) [20]. Likewise, a VSD study found no association with LBW with an adjusted RR of 0.92 (0.78–1.09) [19]. In contrast, the Texan study found greater risk for lower birthweights among decliners for Tdap in the third, fifth and tenth percentiles (*p* = 0.004, 0.002, and 0.032, respectively) [21]. We also had insufficient observations to measure risk for stillbirth among our cohort of infants exposed to maternal Tdap, although other cohort studies [21,22] from Texas, USA and the UK have not found any increased risk.

We found a significant association with reduction in SGA, in line with a maternal influenza study in which the protective effect remained even after consideration of time-dependent biases and confounding from baseline [23]. Unlike preterm birth, for which the protective effect of the vaccine disappeared after adjustments, the associations between vaccination and SGA remained consistent in all analytical approaches [23]. Previous published studies have not shown any association between maternal Tdap vaccination and SGA [18,19,20]. Our finding of no association between Tdap and 5-min Apgar score is consistent with other studies [18,22].

While we examined many birth defects, all but two were too rare for analysis. Other studies have not identified any increased risk for birth defects associated with maternal Tdap [20,21,24]. Due to the increased cases of microcephaly reported in Brazil and their temporal association with the recommendations for maternal Tdap, we specifically aimed to assess this as an outcome. However, we had insufficient observations, with none reported in the vaccine-exposed group and only three in the un-exposed group.

While most of our outcomes had a reduced risk or no association with maternal Tdap, we did find increased odds of ankyloglossia and neonatal erythema toxicum diagnoses among infants born to vaccinated mothers. Both are likely a result of residual confounding, or spurious association to the large number of endpoints. In NZ, both the diagnosis and management of ankyloglossia is controversial with opposing views on the need for treatment and a strong link to the diagnosis and management approaches for lactation disorders. The reported incidence in NZ has increased more than fivefold between 2007 and 2013 with variability in rates of diagnosis and management by region, ethnicity and socioeconomic group. This suggests an inconsistent diagnostic approach, which therefore impacts the reliability of these results. Erythema toxicum is a common rash in neonates and a diagnosis is strongly linked to health-seeking behaviour. We did not consider either of these outcomes to be related to maternal Tdap.

While we adjusted for important confounding variables, including maternal age, ethnicity, socioeconomic status, ANC history, BMI, history of chronic disease, and parity, there may be residual confounding due to important variables not being included in administrative health datasets, such as maternal educational level or other provider/patient characteristics. For example, provider recommendation is an important predictor of a woman receiving Tdap vaccination during pregnancy [25]. Providers that recommend Tdap vaccination during pregnancy are likely to have other differences, such as the characteristics of their patients, type of patient selection and patient care. Measurement error and misclassification of binary confounders can also contribute to residual confounding. Further, we examined many exposures and did not consider confounders on an outcome-by-outcome basis. This analysis approach may have contributed to residual confounding leading to biased estimates.

There are other limitations of health administrative datasets. The NZ National Minimum Data Set is limited to hospital inpatient diagnostic codes for which the validity cannot be assessed, with the risk of a false positive and bias towards the null hypothesis; as such, the incidence rates of adverse infant outcomes may be underestimated. Additionally, a hospitalization diagnosis code does not necessarily reflect an incident outcome, as some outcomes may have occurred or presented earlier in pregnancy and only present in later pregnancy with severity requiring hospitalization. As previously reported [15], we conducted a small validation study of the Tdap exposure variable, comparing primary healthcare organisation (PHO) data and found the immunisation dataset Tdap exposure had high specificity (98.8–99.7%), but low sensitivity (9–61%) among 22,710 pregnant women across seven PHOs, indicating 64% of pregnant women receiving Tdap were incorrectly classified as unvaccinated in the immunisation dataset. In the current study, differential misclassification of the Tdap exposure could be caused by differential quality of data across PHOs and hospitals and could lead to either over- or underestimation of the effects of Tdap on neonatal outcomes. This potential exposure misclassification means that study results should be cautiously interpreted. The small validation study comparing information from the NIR and DHB regarding Tdap exposure was limited to the regional DHB database available at the time of analysis. As this was not nationally representative, a sensitivity analysis using DHB-level data would be inconclusive and could not be directly compared with the national results using the NIR database. In the current study, we only examined data for the first year after implementation (2013), which led to small numbers of outcomes. In addition, we did not account for infant primary vaccinations in our analysis for the longer-term follow-up period. However, the current approach will allow for repeated analyses in future years using the same databases, which are expected to improve over time.

## 6. Conclusions

Results from this study of adverse outcomes following exposure to maternal Tdap vaccination among the infants of pregnant women in NZ are consistent with other studies and provide further support for the safety of the Tdap vaccination during pregnancy. This study evaluated a comprehensive range of infant outcomes in a national population cohort with up to one year follow up. Our findings support the safety of administration of pertussis immunisation during pregnancy.

## Figures and Tables

**Figure 1 vaccines-07-00147-f001:**
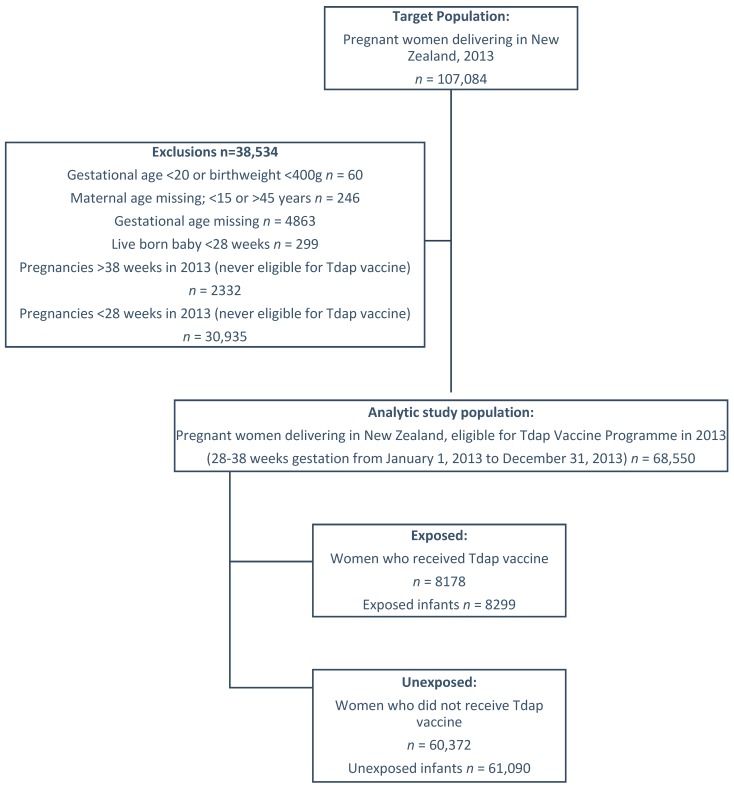
Flow of study population.

**Table 1 vaccines-07-00147-t001:** Demographics of infants born to women * who were eligible † to receive funded vaccination during pregnancy between 1 January and 31 December 2013, New Zealand.

	Mother Tdap Vaccination				Total	
	Infants Exposed		Infants Unexposed			
	N	%	N	%	N	%
Total	8299	12.0	61,090	88.0	69,389	100.0
**Infant characteristics**						
Gender						
Male	4249	51.2	31,283	51.2	35,532	51.2
Female	4050	48.8	29,805	48.8	33,855	48.8
Missing	0	0.0	2	0.0	2	0.0
Infant ethnicity						
Maori	1098	13.2	17,271	28.3	18,369	26.5
Pacific	420	5.1	7022	11.5	7442	10.7
Asian	1216	14.7	8629	14.1	9845	14.2
European/Other	5563	67.0	28,142	46.1	33,705	48.6
Missing	2	0.0	26	0.0	28	0.0
NZ Deprivation Index 2013						
1–2 (least deprived)	1805	21.7	7937	13.0	9742	14.0
3–4	1783	21.5	9216	15.1	10,999	15.9
5–6	1697	20.4	10,861	17.8	12,558	18.1
7–8	1751	21.1	14,106	23.1	15,857	22.9
9–10 (most deprived)	1263	15.2	18,963	31.0	20,226	29.1
Missing	0	0.0	7	0.0	7	0.0
Apgar score at 5 min after birth						
Mean (SD)	9.5 (0.9)		9.5 (0.9)		9.5 (0.9)	
Median (Q1,Q3)	10.0 (9.0,10.0)		10.0 (9.0,10.0)		10.0 (9.0,10.0)	
Birth weight (g)						
Mean (SD)	3467.3 (532.5)		3429.0 (592.5)		3433.6 (585.6)	
Median (Q1,Q3)	3485.0 (3140.0,3800.0)		3450.0 (3090.0,3795.0)		3450.0 (3100.0,3800.0)	
**Mother characteristics**						
Age (years) at last menstrual period						
Mean (SD)	30.7 (5.4)		28.3 (6.1)		28.6 (6.1)	
Median (Q1,Q3)	31.0 (27.0,35.0)		28.0 (24.0,33.0)		29.0 (24.0,33.0)	
Parity						
0	3642	43.9	21,916	35.9	25,558	36.8
1	2915	35.1	19,085	31.2	22,000	31.7
2	1023	12.3	8840	14.5	9863	14.2
3	266	3.2	3652	6.0	3918	5.6
4	71	0.9	1529	2.5	1600	2.3
5	25	0.3	783	1.3	808	1.2
6+	16	0.2	752	1.2	768	1.1
Missing	341	4.1	4533	7.4	4874	7.0
History of stillbirth						
Yes	77	0.9	596	1.0	673	1.0
No	8222	99.1	60,494	99.0	68,716	99.0
History of preterm birth						
Yes	214	2.6	1961	3.2	2175	3.1
No	8085	97.4	59,129	96.8	67,214	96.9
History of chronic disease						
Yes	111	1.3	956	1.6	1067	1.5
No	8188	98.7	60,134	98.4	68,322	98.5
History of antenatal care (no. of LMC visits)						
Mean (SD)	9.7 (3.5)		9.2 (3.8)		9.3 (3.8)	
Median (Q1,Q3)	10.0 (7.0,12.0)		9.0 (7.0,12.0)		9.0 (7.0,12.0)	
Model of care						
DHB	325	3.9	3388	5.5	3713	5.4
MWF	6615	79.7	51,181	83.8	57,796	83.3
GP	111	1.3	442	0.7	553	0.8
OBS	1024	12.3	2878	4.7	3902	5.6
Other	13	0.2	28	0.0	41	0.1
No LMC	211	2.5	3173	5.2	3384	4.9
Mother BMI						
Mean (SD)	25.4 (5.4)		26.5 (6.2)		26.3 (6.1)	
Median (Q1,Q3)	24.0 (22.0,28.0)		25.0 (22.0,30.0)		25.0 (22.0,29.0)	
Current tobacco use						
Yes	550	6.6	11,823	19.4	12,373	17.8
No	7749	93.4	49,267	80.6	57,016	82.2
Influenza vaccination						
Yes	3833	46.2	5187	8.5	9020	13.0
No	4466	53.8	55,903	91.5	60,369	87.0

* Women with a surviving fetus at 20 weeks gestation or who delivered an infant weighing at least 400 g. † 28–38 weeks gestation during 2013. Notes: SD = standard deviation; Q1 = first quartile; Q3 = third quartile; DHB = district health board; MWF = midwife; GP = general practitioner; OBS = obstetrician; LMC = lead maternity carer; BMI = body mass index.

**Table 2 vaccines-07-00147-t002:** Effect of maternal Tdap on hospital-related infant outcomes diagnosed at birth, on eligible maternities, * by Tdap exposure, New Zealand (N_Exposed_ = 8299; N_Unexposed_ = 61,090; N_Total_ = 69,389).

Outcome †	Description	Tdap (Exposed = 1, Unexposed = 0)	N (%)	Unadjusted OR ‡ (95% CI)	*p* Value	Adjusted OR ‡§ (95% CI)	*p* Value
Effects of delivery							
P00	Fetus and newborn affected by maternal conditions	1	14 (0.2)	1.085 (0.619,1.902)	0.7760	1.062 (0.558,2.020)	0.8550
		0	95 (0.2)				
P01	Fetus and newborn affected by maternal complications of pregnancy	1	19 (0.2)	0.999 (0.618,1.614)	0.9968	0.906 (0.526,1.561)	0.7229
		0	140 (0.2)				
P02	Fetus and newborn affected by abnormality of membranes	1	27 (0.3)	1.163 (0.774,1.746)	0.4670	0.911 (0.557,1.490)	0.7112
		0	171 (0.3)				
P03	Fetus and newborn affected by complications of labor and delivery	1	131 (1.6)	1.169 (0.971,1.407)	0.0999	0.931 (0.750,1.155)	0.5141
		0	827 (1.4)				
Physical examination and anthropometric measurements							
P05.12	Small for gestational age (SGA)	1	117 (1.4)	0.711 (0.587,0.861)	**0.0005**	0.721 (0.574,0.905)	**0.0047**
		0	1204 (2.0)				
P05.29	Other fetal malnutrition	1	92 (1.1)	0.832 (0.670,1.034)	0.0969	1.036 (0.807,1.329)	0.7833
		0	812 (1.3)				
P07.13	Low birth weight (LBW): 1500 to <2500 g	1	186 (2.2)	0.755 (0.648,0.880)	**0.0003**	0.784 (0.653,0.941)	**0.0089**
		0	1800 (2.9)				
P07.323	Moderate to late preterm: 32 to <37 weeks	1	398 (4.8)	0.852 (0.766,0.947)	**0.0031**	0.831 (0.729,0.947)	**0.0055**
		0	3412 (5.6)				
P08.0	High birth weight	1	45 (0.5)	0.883 (0.647,1.204)	0.4303	1.157 (0.824,1.625)	0.3995
		0	375 (0.6)				
P08.1	Large for gestational age infants	1	31 (0.4)	0.680 (0.470,0.983)	**0.0403**	0.567 (0.359,0.894)	**0.0147**
		0	335 (0.5)				
P12	Scalp injury due to birth trauma	1	84 (1.0)	1.171 (0.929,1.475)	0.1819	0.940 (0.720,1.228)	0.6514
		0	529 (0.9)				
P15.4	Birth trauma to face	1	28 (0.3)	1.079 (0.726,1.606)	0.7063	0.772 (0.484,1.232)	0.2781
		0	191 (0.3)				
P15.8	Other specified birth trauma	1	10 (0.1)	0.775 (0.404,1.487)	0.4430	0.670 (0.308,1.458)	0.3122
		0	95 (0.2)				
Neonatal conditions classified by organ system							
P20	Intrauterine hypoxia	1	28 (0.3)	0.698 (0.473,1.029)	0.0692	0.670 (0.429,1.047)	0.0788
		0	295 (0.5)				
P21	Asphyxia	1	49 (0.6)	1.135 (0.839,1.535)	0.4104	1.374 (0.968,1.951)	0.0751
		0	318 (0.5)				
P22.0	Respiratory distress syndrome	1	121 (1.5)	0.635 (0.526,0.765)	**<0.0001**	0.652 (0.524,0.811)	**0.0001**
		0	1392 (2.3)				
P22.1	Transient tachypnea of newborn	1	247 (3.0)	0.891 (0.779,1.018)	0.0906	0.839 (0.721,0.975)	**0.0224**
		0	2034 (3.3)				
P22.89_P28.2589	Respiratory distress	1	230 (2.8)	1.060 (0.921,1.219)	0.4165	0.998 (0.850,1.171)	0.9775
		0	1600 (2.6)				
P23	Congenital pneumonia	1	24 (0.3)	0.870 (0.569,1.329)	0.5192	1.010 (0.629,1.622)	0.9675
		0	203 (0.3)				
P24.0	Meconium aspiration syndrome	1	11 (0.1)	0.736 (0.396,1.369)	0.3330	0.977 (0.497,1.921)	0.9461
		0	110 (0.2)				
P25	Interstitial emphysema and related conditions	1	43 (0.5)	1.344 (0.970,1.861)	0.0754	1.240 (0.860,1.787)	0.2495
		0	236 (0.4)				
P28.34	Apnea	1	70 (0.8)	0.702 (0.549,0.899)	**0.0049**	0.777 (0.585,1.031)	0.0803
		0	731 (1.2)				
P29.1	Tachycardia or bradycardia	1	55 (0.7)	0.785 (0.594,1.037)	0.0888	0.691 (0.501,0.954)	**0.0245**
		0	515 (0.8)				
P29.82	Benign and innocent cardiac murmurs in newborn	1	34 (0.4)	1.193 (0.830,1.715)	0.3415	1.098 (0.724,1.667)	0.6598
		0	210 (0.3)				
P36.89	Bacterial sepsis of newborn, specified or unspecified	1	37 (0.4)	0.803 (0.571,1.128)	0.2052	0.872 (0.599,1.270)	0.4763
		0	339 (0.6)				
P37.5	Candidiasis	1	17 (0.2)	0.702 (0.427,1.156)	0.1645	0.656 (0.381,1.131)	0.1294
		0	178 (0.3)				
P38	Omphalitis	1	29 (0.3)	1.180 (0.797,1.748)	0.4085	1.399 (0.875,2.236)	0.1606
		0	181 (0.3)				
P39.1	Neonatal conjunctivitis and dacryocystitis	1	82 (1.0)	0.961 (0.763,1.211)	0.7383	0.841 (0.641,1.103)	0.2111
		0	628 (1.0)				
P39.4	Neonatal skin infection	1	13 (0.2)	1.153 (0.642,2.070)	0.6329	1.352 (0.706,2.593)	0.3631
		0	83 (0.1)				
P54	Neonatal hemorrhage	1	18 (0.2)	0.914 (0.560,1.492)	0.7180	0.981 (0.571,1.683)	0.9435
		0	145 (0.2)				
P55	Haemolytic diseases	1	39 (0.5)	0.689 (0.496,0.957)	**0.0263**	0.663 (0.444,0.990)	**0.0445**
		0	416 (0.7)				
P59	Other neonatal jaundice	1	308 (3.7)	0.827 (0.733,0.932)	**0.0019**	0.869 (0.757,0.998)	**0.0466**
		0	2722 (4.5)				
P61.0	Thrombocytopenia	1	13 (0.2)	0.633 (0.359,1.116)	0.1141	0.830 (0.440,1.567)	0.5657
		0	151 (0.2)				
P61.234	Anaemia	1	20 (0.2)	0.574 (0.364,0.905)	**0.0170**	0.461 (0.270,0.786)	**0.0045**
		0	256 (0.4)				
P70.0	Syndrome of infant of mother with gestational diabetes	1	50 (0.6)	0.682 (0.510,0.912)	**0.0099**	0.683 (0.487,0.960)	**0.0281**
		0	538 (0.9)				
P70.1	Syndrome of infant of a diabetic mother	1	18 (0.2)	0.633 (0.391,1.025)	0.0631	0.601 (0.278,1.300)	0.1957
		0	209 (0.3)				
P70.34	Hypoglycemia	1	236 (2.8)	0.792 (0.691,0.907)	**0.0008**	0.795 (0.681,0.929)	**0.0038**
		0	2178 (3.6)				
P74.1	Dehydration of newborn	1	58 (0.7)	1.155 (0.875,1.525)	0.3091	1.093 (0.794,1.503)	0.5868
		0	370 (0.6)				
P74.23	Electrolyte anomalies (Na, K)	1	57 (0.7)	0.781 (0.594,1.028)	0.0778	0.844 (0.621,1.147)	0.2786
		0	536 (0.9)				
P80	Hypothermia	1	73 (0.9)	0.861 (0.675,1.099)	0.2297	0.964 (0.726,1.279)	0.7996
		0	623 (1.0)				
P81	Other disturbances of temperature regulation of newborn	1	59 (0.7)	1.287 (0.975,1.699)	0.0744	1.297 (0.947,1.775)	0.1052
		0	338 (0.6)				
P83.1	Neonatal erythema toxicum	1	47 (0.6)	1.583 (1.154,2.171)	**0.0044**	1.661 (1.163,2.372)	**0.0052**
		0	219 (0.4)				
P83.5	Congenital hydrocele	1	11 (0.1)	0.771 (0.414,1.436)	0.4123	0.782 (0.396,1.543)	0.4779
		0	105 (0.2)				
P83.89	Other conditions of integument	1	20 (0.2)	0.909 (0.571,1.447)	0.6862	0.986 (0.594,1.637)	0.9572
		0	162 (0.3)				
P90	Seizure	1	18 (0.2)	0.974 (0.596,1.594)	0.9179	1.059 (0.602,1.862)	0.8422
		0	136 (0.2)				
P91.6	Hypoxic ischemic encephalopathy	1	12 (0.1)	0.874 (0.480,1.592)	0.6606	0.786 (0.390,1.585)	0.5011
		0	101 (0.2)				
P92.0	Vomiting	1	36 (0.4)	1.228 (0.862,1.749)	0.2549	0.928 (0.612,1.406)	0.7231
		0	216 (0.4)				
P92.123589	Difficulty feeding	1	369 (4.4)	1.239 (1.107,1.386)	**0.0002**	1.054 (0.924,1.203)	0.4344
		0	2212 (3.6)				
P94.2	Congenital hypotonia	1	15 (0.2)	0.898 (0.525,1.535)	0.6929	0.788 (0.413,1.504)	0.4703
		0	123 (0.2)				
P96.81	Jittery baby	1	32 (0.4)	1.197 (0.823,1.740)	0.3473	1.104 (0.705,1.728)	0.6666
		0	197 (0.3)				
Congenital anomalies							
Q38.1	Ankyloglossia	1	221 (2.7)	1.545 (1.334,1.789)	**<0.0001**	1.241 (1.044,1.474)	**0.0143**
		0	1063 (1.7)				
Q66	Talipes equinovarus, metatarsus varus, or other congenital deformities of feet	1	33 (0.4)	0.880 (0.613,1.263)	0.4873	0.963 (0.612,1.516)	0.8707
		0	276 (0.5)				

* 28–38 weeks gestation during 2013. † Table includes outcomes diagnosed at delivery where N ≥ 10 and % ≥ 0.1 (please see for ICD-10-AM code map). ‡ OR = odds ratio, which compares mothers exposed to Tdap with those unexposed. OR > 1 indicates greater likelihood of exposed group having the outcome if *p*-value < 0.05. § Logistic regression model adjusted for birth status (single live birth, other birth); maternal ethnicity (Maori, Pacific, Asian, NZ European or other); NZ Deprivation Index 2013 (1–10); maternal age (in years); history of antenatal care (total no. of lead maternity carer visits); body mass index (kg/m^2^); history of chronic disease (yes, no); parity (0, 1+); model of care (DHB, midwife, obstetrician/general practitioner, no lead maternity carer/other); and influenza vaccination (yes, no). Notes: Q1 = first quartile; Q3 = third quartile; CI = confidence interval; ICD-10-AM = International Classification of Diseases, Tenth Revision, Australian Modification. Bold denotes statistically significant outcomes.

**Table 3 vaccines-07-00147-t003:** Effect of maternal Tdap on infant’s Apgar score at 5 min after birth and birthweight at delivery, on eligible maternities, * by Tdap exposure, New Zealand (N_Exposed_ = 8299; N_Unexposed_ = 61,090; N_Total_ = 69,389).

Outcome †	Tdap	N	Mean (SD)	Unadjusted Mean Difference (95% CI)	*p* Value	Adjusted Mean Difference (95% CI)	*p* Value
Apgar score at 5 min after birth	Exposed	7660	9.537 (0.879)	0.005 (−0.015,0.026)	0.6246	0.000 (−0.022,0.023)	0.9775
	Unexposed	53,810	9.531 (0.870)				
Birthweight (g)	Exposed	8063	3467 (532)	38.275 (24.640,51.911)	**<0.0001**	35.585 (21.392,49.778)	**<0.0001**
	Unexposed	58,337	3429 (592)				

* 28–38 weeks gestation during 2013. † Generalized linear model adjusted for birth status (single live birth, other birth); maternal ethnicity (Maori, Pacific, Asian, NZ European or other); NZ Deprivation Index 2013 (1–10); maternal age (in years); history of antenatal care (total no. of lead maternity carer visits); body mass index (kg/m^2^); history of chronic disease (yes, no); parity (0, 1+); model of care (DHB, midwife, obstetrician/general practitioner, no lead maternity carer/other); and influenza vaccination (yes, no). Notes: SD = standard deviation; CI = confidence interval. Bold denotes statistically significant outcomes.

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
