# Peer review of "A Retrospective Cohort Study of Safety Outcomes in New Zealand Infants Exposed to Tdap Vaccine in Utero"

_vaccines, 2019, doi:10.3390/vaccines7040147_

Round 1
Reviewer 1 Report
General Comments
This is a well-written and interesting manuscript that adds to the literature. The study appears to be well-done with acceptable research methods used. The paper states in several places that “there was no association between maternal Tdap and stillbirth”, but also states that “There were insufficient number of cases in the vaccine-exposed group to assess the association between Tdap exposure and stillbirth (n=9)”. Therefore, all statements that there is no association between maternal Tdap and stillbirth need to be removed. This statement cannot be made if there were insufficient cases to analyze the association. See the Abstract Line 35, Discussion Line 256, 272, and please check the text carefully for other instances. The same statement is also made regarding no association with microcephaly (Line 257) although there were insufficient number of microcephaly cases for analysis. Please check for additional instances of this throughout the paper, and remove all statements of non-association when there were insufficient numbers of cases for analysis. Inclusion criteria for the study was based on eligibility for the government funded, national-level vaccination program. Perhaps this is clear to readers in NZ, but not to other international readers; what proportion of the NZ population is eligible for this government funded vaccination program? I suggest adding a sentence to the first paragraph of the Methods describing who is/is not eligible, and roughly what proportion of pregnant women in NZ are eligible. Do the authors think that women eligible for the government funded vaccination program are representative of all pregnant women in NZ? The two main limitations of the paper are the potential for misclassification of the outcomes and almost certain misclassification of the exposure. Since the outcomes were defined based only on hospitalization data, and many of these outcomes may not require hospitalization but only out-patient care, the outcomes are likely underestimated, as the authors stated. The authors’ validation study found that up to 64% of pregnant women receiving Tdap were incorrectly classified as unvaccinated in the immunisation dataset. The authors should conduct sensitivity analyses to assess the potential for bias associated with these misclassified variables. Figure 1 shows the study population as it relates to the numbers of pregnant women, but since all analyses and tables are based on number of infants, the authors should consider presenting Figure 1 as it relates to the numbers of infants instead.
Specific Comments
Lines 90-91: Please clarify whether the independent variable was Tdap during the pregnancy at any time, or only during the recommended time (28-38 weeks). For example, if a mother was vaccinated at 6 weeks gestation, was that considered an exposure or not? Lines 166-168: The authors provide demographics percents for a few of the demographic variables. Instead of presenting the percents for the vaccine-exposed group, this section should describe the total study sample and present the demographics for the total sample, rather than just one or the other of the exposed/unexposed groups. Table 1 is very large (3.5 pages) and should be condensed. Select the most important demographics to include in the table; could think about discarding the geographic areas, for the continuous variables showing either the mean/median or the categories but not both, and discarding some of the other secondary demographics/health outcomes that are shown elsewhere or of lesser importance. Table 2: Since there are many outcomes described in this table, it would be helpful if the outcomes are organized in the same way that they are discussed in the Results section. I.e., organize the outcomes by the categories Events of delivery, Physical examination and anthropometric measurements, Congenital anomalies, etc. Most of the numbers presented in the Results section have 2 decimal places, but a few have 3 decimal places (e.g., Lines 215, 220, 221, etc.). The numbers of decimal places should be consistent throughout the text. Lines 221-222: This is the first place where the term “restricted cohort” is used. Is the restricted cohort the full study sample? Or is it some further restricted cohort? If the authors want to use this term, they should introduce it and define it earlier in the Methods section. However, if the “restricted cohort” is the same as the total study sample (i.e., all 69,389 infants included in the study sample), then I suggest dropping the use of the term “restricted cohort”.
Author Response
The authors would like to thank the reviewers for their time and helpful feedback. We have addressed the comments to reviewer 1 below in bold.
The paper states in several places that “there was no association between maternal Tdap and stillbirth”, but also states that “There were insufficient number of cases in the vaccine-exposed group to assess the association between Tdap exposure and stillbirth (n=9)”. Therefore, all statements that there is no association between maternal Tdap and stillbirth need to be removed. This statement cannot be made if there were insufficient cases to analyze the association. See the Abstract Line 35, Discussion Line 256, 272, and please check the text carefully for other instances.
Thankyou, we agree and this has been amended.
The same statement is also made regarding no association with microcephaly (Line 257) although there were insufficient number of microcephaly cases for analysis. Please check for additional instances of this throughout the paper, and remove all statements of non-association when there were insufficient numbers of cases for analysis.
Also amended.
Inclusion criteria for the study was based on eligibility for the government funded, national-level vaccination program. Perhaps this is clear to readers in NZ, but not to other international readers; what proportion of the NZ population is eligible for this government funded vaccination program? I suggest adding a sentence to the first paragraph of the Methods describing who is/is not eligible, and roughly what proportion of pregnant women in NZ are eligible.
We have added ‘all pregnant women in NZ” …between 28-38 weeks gestation.
Do the authors think that women eligible for the government funded vaccination program are representative of all pregnant women in NZ?
As per clarification above – all women are eligible.
The two main limitations of the paper are the potential for misclassification of the outcomes and almost certain misclassification of the exposure. Since the outcomes were defined based only on hospitalization data, and many of these outcomes may not require hospitalization but only out-patient care, the outcomes are likely underestimated, as the authors stated. The authors’ validation study found that up to 64% of pregnant women receiving Tdap were incorrectly classified as unvaccinated in the immunisation dataset. The authors should conduct sensitivity analyses to assess the potential for bias associated with these misclassified variables.
We discussed this as length during the analysis. The small validation study comparing information from the NIR and DHB regarding Tdap exposure was limited to regional DHB database available at the time of analysis. As this was not nationally representative, a sensitivity analysis using DHB-level data would be inconclusive and could not be directly compared with the national results using the NIR database. A better approach we agreed was to collect ongoing national data and repeat the same analysis over time with the expectation that the national database will improve over time.
We have added to the discussion section: “The small validation study comparing information from the NIR and DHB regarding Tdap exposure was limited to regional DHB database available at the time of analysis. As this was not nationally representative, a sensitivity analysis using DHB-level data would be inconclusive and could not be directly compared with the national results using the NIR database.”
Figure 1 shows the study population as it relates to the numbers of pregnant women, but since all analyses and tables are based on number of infants, the authors should consider presenting Figure 1 as it relates to the numbers of infants instead.
We have clarified the headings at the top of the table to make this clearer.
Lines 90-91: Please clarify whether the independent variable was Tdap during the pregnancy at any time, or only during the recommended time (28-38 weeks). For example, if a mother was vaccinated at 6 weeks gestation, was that considered an exposure or not?
Yes, the independent variable is receipt of Tdap during pregnancy. The study cohort included all women who were eligible to receive funded Tdap during the study period, however, the exposure to Tdap included all vaccinations they received during pregnancy. The analysis did take into account all vaccinations received during pregnancy, with the fact that only 5% received outside of the 28-38 gestation window.
We have clarified in the text in the methods section: “Analysis takes in to account all Tdap vaccinations during pregnancy with 5% occurring outside of the 28-38 gestation window.”
Lines 166-168: The authors provide demographics percents for a few of the demographic variables. Instead of presenting the percents for the vaccine-exposed group, this section should describe the total study sample and present the demographics for the total sample, rather than just one or the other of the exposed/unexposed groups.
We think this is clear – the table is the study population and the total is in the last column.
Table 1 is very large (3.5 pages) and should be condensed. Select the most important demographics to include in the table; could think about discarding the geographic areas, for the continuous variables showing either the mean/median or the categories but not both, and discarding some of the other secondary demographics/health outcomes that are shown elsewhere or of lesser importance.
We have trimmed the table.
Table 2: Since there are many outcomes described in this table, it would be helpful if the outcomes are organized in the same way that they are discussed in the Results section. I.e., organize the outcomes by the categories Events of delivery, Physical examination and anthropometric measurements, Congenital anomalies, etc.
Additional headings have been added to the table.
Most of the numbers presented in the Results section have 2 decimal places, but a few have 3 decimal places (e.g., Lines 215, 220, 221, etc.). The numbers of decimal places should be consistent throughout the text.
We have amended, all are now to two decimal places.
Lines 221-222: This is the first place where the term “restricted cohort” is used. Is the restricted cohort the full study sample? Or is it some further restricted cohort? If the authors want to use this term, they should introduce it and define it earlier in the Methods section. However, if the “restricted cohort” is the same as the total study sample (i.e., all 69,389 infants included in the study sample), then I suggest dropping the use of the term “restricted cohort”.
Restricted cohort has been removed.
Reviewer 2 Report
This is an interesting piece of research that evaluates the immunisation in pregnancy safety. However, there are some minor issues that authors should consider before publishing-
The title of the research is too broad, make it concise and more informative. Abstract provides methodological information clearly and the aim of the research is included. In the introduction, line 78, the authors mentioned it is a second paper. What do they mean by second paper? What was the first paper? How it is linked to this paper? If it is not relevant, they should remove it. The only drawback I found in the methods section is that the data is from 2013, not very old but also not fresh. So the editor should decide how this paper will be helpful. Results and discussion sections are well written.Author Response
The authors would like to thank the reviewers for their time and helpful feedback. We have addressed the comments to reviewer 2 below in bold.
The title of the research is too broad, make it concise and more informative.
We have deleted the first part of the title and added New Zealand. It now reads “A retrospective cohort study of safety outcomes in New Zealand infants exposed to Tdap vaccine in utero.”
In the introduction, line 78, the authors mentioned it is a second paper. What do they mean by second paper? What was the first paper? How it is linked to this paper? If it is not relevant, they should remove it.
This has been deleted. It now reads: “We previously reported the maternal outcomes from this study 16; here, we report infant outcomes by examining the difference in birth and hospital-related outcomes of infants with and without fetal exposure to Tdap.”